



# Sectoral contributions of high-emitting methane point sources from major U.S. onshore oil and gas producing basins using airborne measurements from MethaneAIR

Jack D. Warren[*1], Maryann Sargent[2], James P. Williams[1], Mark Omara[1,3], Christopher C. Miller[1,2,3], Sebastien Roche[1,2,3], Katlyn MacKay[1], Ethan Manninen[2], Apisada Chulakadabba[2], Anthony Himmelberger[3], Joshua Benmergui[1,2,3], Zhan Zhang[2], Luis Guanter[1], Steven Wofsy[2], Ritesh Gautam*[1,3]

[1]Environmental Defense Fund, New York, NY, USA 10010

[2]Harvard University, Cambridge, MA, USA 02138

[3]MethaneSAT, LLC, Austin, TX, USA 78701

*Correspondence to*: Jack D. Warren (jwarren@edf.org), Ritesh Gautam (rgautam@edf.org)

**Abstract.** High-emitting methane point sources, quantified by remote sensing methods at individual facilities, have gained

significant interest for enabling rapid monitoring and mitigation of methane emissions from the oil and gas sector. Here, we present new methane point source quantifications from MethaneAIR, the airborne precursor to MethaneSAT, from campaigns in 2021-2023 which targeted major oil and gas basins covering ~80% of U.S. onshore production. Flying at ~12 km above ground, MethaneAIR provides wide-area methane mapping and high-resolution measurements of high-emitting methane point sources. Across 13 major basins, MethaneAIR detected over 400 point sources with emission rates > ~200 kg h[-1], for which

we performed detailed attribution to facility categories within oil and gas and non-oil and gas sectors. In 2023, we quantified total point source methane emissions of 360 t h[-1] (95% confidence interval: 285-445 t h[-1]), with ~80% of the total attributable to oil and gas sources. Non-oil and gas sources made up 50-80% of observed point source emissions in certain basins, with coal facilities in the Appalachian being the largest source of non-oil and gas methane emissions (24-40 t h[-1]). We observe emission source intermittency and significant variation across facility types and basins, highlighting the complex

characteristics of high-emitting point sources. Our results emphasize the importance of detailed source attribution for prioritizing mitigation efforts and provide the first analysis of methane point sources in several regions, which will be improved by the observational capabilities of a growing set of methane satellites.

## 1 Introduction

Methane is a short-lived greenhouse gas responsible for over a quarter of today's warming (Ocko et al., 2018). Multinational

agreements, including the Global Methane Pledge (https://www.globalmethanepledge.org/), have pledged to reduce anthropogenic methane emissions 30% by 2030, with a specific focus on emissions coming from the oil and gas industry that



make up approximately a quarter of anthropogenic methane emissions (IEA, 2024). National inventories of methane emissions, such as the EPA Greenhouse Gas Inventory (EPA, 2024), offer a way for nations to identify dominant sources of methane emissions and prioritize mitigation efforts. However, peer-reviewed measurement-based studies have consistently

found substantial underestimation – generally by a factor of 2× – of the magnitude of total oil and gas methane emissions when compared with official "bottom-up" inventories (Alvarez et al., 2018; Lu et al., 2023; Shen et al., 2022). Previous research suggests that a key contributor to this gap comes from a small number of high-emitting methane point sources (Brandt et al., 2014; Omara et al., 2018; Zavala-Araiza et al., 2017), whose contributions to regional and national methane emissions are not adequately accounted for in "bottom-up" source-level inventory data and methods.

High-emitting oil and gas methane point sources, with individual methane emission rates generally of the order of several tens to thousands of kg h$^{-1}$ methane emissions (Varon et al., 2018; Cusworth et al., 2022; Irakulis-Loitxate et al., 2021), are often the result of intentional and unplanned emission venting due to abnormal process conditions, including equipment malfunctions and blowdowns (Duren et al., 2019; Zavala-Araiza et al., 2017). These emissions can be short in duration (Tullos et al., 2021), but recent research suggests that point source emissions may be persistent or recurring in many cases (Cusworth et al., 2021).

Several studies using airborne spectrometers and LiDAR have demonstrated the ability to quantify methane point sources across several regions in the U.S. (Cusworth et al., 2022; Kunkel et al., 2023; Sherwin et al., 2024). Research by Cusworth et al. (2022) comparing point source quantifications to overlapping satellite-based inversions showed that methane point sources above 10 kg h$^{-1}$ can make up 13-59% of total regional flux in the U.S. Globally, sources with individual facility-level methane emissions greater than approximately 10 t h$^{-1}$, are estimated to contribute 8 to 12% (~8 Tg per year) of total global oil and gas

production methane emissions (Lauvaux et al., 2022).

Previous research of high-emitting methane point sources has indicated variation in the overall contribution from oil and gas industry segments (e.g., upstream versus midstream facilities) from basin to basin (Cusworth et al., 2022). Additionally, large point sources can occur at non-oil and gas facilities, such as landfills (Cusworth et al., 2024) or coal mines (Sadavarte et al., 2022) and can possibly contribute a significant portion of emissions in specific oil and gas basins. However, the investigations

of large point sources by facility types have been limited to a relatively few basins in the U.S. (Cusworth et al., 2022), with most research focusing on the Permian (Chen et al., 2022; Cusworth et al., 2021; Kunkel et al., 2023; Yu et al., 2022), the largest oil producing basin in the U.S. (Enverus, 2024), and where almost all high-emitting methane point source emissions originate from oil and gas sources. Given that operational practices and emission dynamics can change over time (Lyon et al., 2021), there is a strong need to close the geographic and temporal gaps in high-resolution remote sensing measurements of

methane point sources to characterize emissions magnitude and variation across basins.

In this study we use MethaneAIR, an airborne imaging spectrometer with capabilities of quantifying both high-emitting methane point sources and diffuse area emissions, to investigate the trends and magnitude of point source emissions in the U.S. The instrument is the precursor to the MethaneSAT satellite mission and is designed to fly at a 12 km altitude and observe





a 4.5 km swath. The aircraft has much greater spatial coverage than other airborne spectrometers currently used to detect
methane emissions, yet still maintains high enough spatial resolution to enable facility level investigations (Staebell et al.,
2021). We use MethaneAIR's flights from 2021-2023 to explore the contribution of methane emissions from over 400 high-
emitting methane point sources across all major oil and gas regions in the U.S., many of which have been previously unexplored
in the literature. From these novel point source quantifications, we investigate how high-emitting methane point sources are
distributed across facility types, and if any patterns of emissions emerge across different regions with varying mix of oil and
gas and non-oil and gas methane sources. Finally, we compare the observed relative contributions of high-emitting methane
point sources across and within sectors, and leverage estimates from prior studies to explore the temporal variation of point
source emissions. Overall, the results of this work emphasize the importance of detailed methane source attribution for
prioritizing mitigation efforts and provide the first analysis of methane point sources in several key U.S. oil and gas production
regions.

## 2 Methods

### 2.1 An overview of the MethaneAIR instrument and 2021-2023 measurement campaigns

Detailed descriptions of the MethaneAIR instrument's calibration (Conway et al., 2024; Staebell et al., 2021), retrieval methods
(Chan Miller et al., 2024), point source quantification (Chulakadabba et al., 2023), and controlled-release validation (Abbadi
et al., 2024) are explored at length in prior studies. MethaneAIR is an infrared imaging spectrometer which uses a $CO_2$ proxy
retrieval to calculate column $xCH_4$. In 2021 and 2022, the MethaneAIR instrument was flown aboard the NCAR GV aircraft
over a few select oil and gas regions (Permian, Uinta, Anadarko). In 2023, the instrument was flown aboard a modified Learjet
aircraft and collected observations for 64 flight days between May 25th and October 15th. MethaneAIR flew over all major
U.S. onshore oil and gas regions (Figure 1), with a goal of observing at least 80% of onshore production within a single year.

Flown at cruising altitude of approximately 12 km, MethaneAIR has a large 4.5 km swath and can cover over 100 km x 100
km in approximately 3 hours. It produces images with 10 m × 10 m resolution with 17-20 ppb standard deviation on a flat
scene (Chulakadabba et al., 2023). As discussed in Chen, Sherwin et al. 2023 and Chen et al., 2024, a single expansive scan
of an entire region to estimate total point source emissions, as is the case with MethaneAIR, will have a lower standard error
than combining several smaller scans for estimating a region's average point source emissions. However, differences in the
facility composition and emission sources in sub-basin regions can increase the required sampling to develop a representative
estimate of basin-level emissions (Chen et al., 2024). Determining the number of overflights needed for a representative sample
is likely possible with the combination of multiscale measurements and long-term campaigns, which is outside the scope of
this work. Therefore, we prioritized our sampling based on the goal of first scanning at least 80% of onshore oil and gas
production, then collecting additional overpasses where conditions were favorable. We then prioritized regions of concentrated
oil and gas activity or where prior investigations occurred for additional overflights.



## 2.2 Methane point source plume identification and flux quantification methods

Within the ~10,000 km$^2$ area xCH$_4$ maps produced by MethaneAIR, coherent plumes were identified using a thresholding method and quantified using a divergence integral (DI) method (Chulakadabba et al., 2023). The automated plume-finding
algorithm uses a two-part threshold-based clumping technique and a manual QA/QC of the found plumes. We first produced a gridded flux product by calculating the divergence of the flux for 600 m × 600 m squares which were tiled across the scene, oversampling by moving the squares over by 200 m at a time (Figure S1.1). The methane flux from each square was calculated using HRRR wind products and the DI method described in (Chulakadabba et al., 2023), and briefly explained in the supplemental material (section 1). We found that the gridded flux product for the scene helped to identify the upwind end of
plumes, which had larger flux divergence than the downwind end of plumes. To locate plumes in the scene, we first took the absolute value of the gridded flux map, as inaccuracies in the meteorological product's wind direction often lead to positive and negative dipoles around plumes. We identified and isolated "clumps" of elevated [absolute value of] flux by defining a threshold of 1.3*[standard deviation of scene's gridded flux] + [mean of scene's gridded flux], and setting values below that threshold to null values. For each flux clump with >12 (200 m x 200 m) pixels, we cropped the xCH$_4$ map to ± 3 km around
the center of the clump. For the cropped scene, we then repeated the thresholding and clumping algorithm to find xCH$_4$ clumps above the threshold. To keep an xCH$_4$ clump, it must have >200 (10 m × 10 m) pixels and part of the clump must be within 1.5 km of the DI clump centroid. Multiple xCH$_4$ clumps can be considered part of the same plume mask if they meet those criteria.

We then calculated the major axis and eccentricity of the xCH$_4$ plume mask. Using the wind direction from HRRR, we found the ends of the xCH$_4$ mask in the upwind and downwind directions, and the difference between the 2 points was taken to be the plume length. The origin of the plume was taken to be either the upwind end of the plume mask, or the center of the elevated DI flux clump, whichever was farther upwind (Figure S1.2). Finally, the flux from the plume was calculated using the growing box DI method, starting at the plume origin, and extending to the plume length, as described by Chulakadabba et
al., 2023 (supplement section 1). Plumes with a flux less than 150 kg h$^{-1}$ were discarded as being below the detection threshold of the methodology. In addition, we manually reviewed all identified plumes and discarded plumes with known artifacts.

## 2.3 Facility attribution of high-emitting methane point sources

Attribution of point sources to facility types has been achieved through spatially querying known infrastructure locations
(Hmiel et al., 2023), or by manual review of available high-resolution satellite imagery (Cusworth et al., 2022; Irakulis-Loitxate et al., 2021). Here, we apply both methods, using a combination of automated spatial querying from a collection of public geospatial oil and gas and non-oil and gas infrastructure datasets with subsequent manual review to identify emitting facility





types. The applied infrastructure collection included state and federal inventories of air emission sources, oil and gas infrastructure databases (Omara et al., 2023), and several sources dedicated to accounting for non-oil and gas methane emitting
sectors such as waste management or concentrated animal feeding operations (CAFOs). Full details on the spatial attribution methods, definitions of facility types, and infrastructure databases used can be found in the Supporting Information (S2).

### 2.4 Analyses of high-emitting methane point source emissions

To calculate a basin's total point source emissions, we divide each plume's quantified emission rate by the total number of overpasses of that respective location before summing all plumes in the basin. This is analogous to persistence-weighted
emissions as described in Chen et al., 2022 and Cusworth et al., 2022. Prior studies have taken an additional step and used Monte Carlo simulations to sample a range of persistence values from a given basin according to the facility type for regions with limited overflights. This approach relies on having a large collection of site level observations that altogether form a representative distribution of a facility's possible intermittency. However, basin and emissions dynamics change over time and samples from prior years may not be representative of recent emissions. Additionally, the intermittency of a facility type may
differ across regions, and since we sampled several regions not previously explored, in many cases we do not have suitable collection of source-level persistence values to sample for persistence simulations. Nonetheless, we explore possible differences of results based on point source total calculation methods (Supporting information S4), in which results are broadly consistent regardless of whether persistence is modelled for subregions with few overflights. For aggregated basin-level point source emissions estimates, we report a 95% confidence interval that is estimated through a simulation-based approach. In
each simulation, we iteratively assign a new emission rate to the full suite of point source detections using a random draw from a normal probability distribution defined by the quantified plume-level emission rate and uncertainty. New emission rates are then divided by the number of overflights and summed per simulation. Reported confidence intervals represent the range between the 2.5th and 97.5th percentile of simulated sums across 10,000 iterations.

Finally, we compare the relative contribution of emissions from various facility types per basin by normalizing total attributed emissions by the basin's total emissions for MethaneAIR's 2023 campaign. To create a definite comparison, we limit this portion of the analysis to emission rates with a high probability of detection. Detection of a point source is dependent on both the size of the source and in-situ conditions, such as wind speed, surface brightness and heterogeneity, or background concentration of methane (Conrad et al., 2023). As the probability of detection decreases, collective point source observations
are not likely to characterize the complete nature of emissions in an area due to the possibility of present but unidentified emission sources in an individual scan. MethaneAIR has been tested in blind controlled release experiments and accurately quantified emissions as low as 33 kg h$^{-1}$ (Abbadi et al., 2024), however the ability to quantify point sources in relatively controlled field testing can greatly differ from basin-scale campaigns where source locations are not known beforehand (Conrad et al., 2023). We define our emission rate threshold for comparison based on the drop-off in observational frequency
in the cumulative observed emissions distribution, which occurs at 550 kg h$^{-1}$ (Supporting Information S5). We then apply this





same detection threshold in overlapping regions for the observations from Cusworth et al. 2022 and MethaneAIR's research flight phase to explore if the relative point source emissions characteristics have changed over time (Chen et al., 2024).

## 3 Results and Discussion

### 3.1 Basin-level and national point source methane emissions and sectoral attribution

Throughout the entire 2023 campaign, MethaneAIR quantified a total of 323 plumes at 268 unique facilities across 13 oil and gas producing basins (Figure 1). Altogether, the 2023 flights in this study comprises a unique area covering 79% of onshore U.S. oil and gas production. To our knowledge, this campaign represents the largest coverage of unique U.S. onshore oil and gas production in a single year by an airborne spectrometer quantifying methane point sources to date. With the addition of the research flights from 2021 and 2022 included in this study, the totals rise to 426 plumes and 80% of onshore production.

MethaneAIR detected and quantified each high-emitting methane point source with individual emission rates $> \sim 200$ kg h$^{-1}$ to ~70 t h$^{-1}$. Detection frequency of plumes peaked at approximately 550 kg h$^{-1}$, which we use as the threshold for subsequent comparisons of sectoral variation (see Supporting information S4 observational frequency and detection limits).

Overall, MethaneAIR quantified an average total of 363 t h$^{-1}$ (95% CI: 285–445 t h$^{-1}$) from high-emitting point sources across

all surveyed regions in 2023, with 290 t h$^{-1}$ (95% CI: 213–366), or ~80% of the total, coming from oil and gas sources (Table 1). The quantified emissions from these MethaneAIR-detected high-emitting point sources represent roughly one-fifth of the estimated national oil and gas methane emissions of ~13 Tg yr$^{-1}$ (~1,500 t h$^{-1}$; Alvarez et al., 2018; Shen et al., 2022).

Point source attribution was successful in specifically determining the facility type of 400 point sources (94%) (Table 1).

MethaneAIR observed emissions from several other facilities not commonly considered for their methane emissions, including power plants, a biogas storage facility, and a fertilizer plant. Across all regions, we identified high-emitting methane point sources at major oil and gas facilities (well sites, natural gas compressor stations, central tank batteries, processing plants, and pipelines) and non-oil and gas facilities (e.g., coal and landfills).

Comparing the relative distributions of emission rates across sectors (Figure 2A), our results indicate some stratification of emission rates between the major oil and gas and non-oil and gas facility types. For oil and gas facilities, the observed median emission rate is consistent across pipelines, well sites, and compressor stations (1.1-0.99 t h$^{-1}$) and slightly elevated at processing plants (1.3 t h$^{-1}$), but all share large amounts of overlap in the interquartile range. Coal facilities had the highest median emission rate (2.0 t h$^{-1}$) and differentiated themselves from all other non-oil and gas facility types which had lower

median emission rates compared to all oil and gas facilities (1.0-0.80 t h$^{-1}$). Due to the abundance of coal sources, the Appalachian-central had the highest median emission rate (1.5 t h$^{-1}$) compared to all other basins (Figure 3A). Conversely,







**Figure 1.** High-emitting methane point sources detected by MethaneAIR from 2021-2023. Emission sources (circles) are sized by emission rate and colored according to emission source type (orange: oil and gas sources; dark green: non-oil and gas sources). Bold black outlines indicate basin boundaries (https://www.eia.gov/maps/maps.php) while gray outlines indicate regions flown by MethaneAIR without the inbound and outbound flight tails. Subpanels depict two examples of the detected methane plumes, labelled with emission rate and basin, from an coal facility (left) and processing plant (right). Discussed geologic sub-basin boundaries are illustrated in the Supporting Information (Figure S3.1).






**Table 1**. Observational summary of MethaneAIR campaigns 2021-2023. Values in parenthesis represent the simulated 95% confidence interval. Total point source emissions are calculated using the persistence-weighted total of all detections.

| | Dates surveyed (MM/DD) | Flights | Detected plumes | Unique emitting facilities | Total point source emissions | | |
|---|---|---|---|---|---|---|---|
| | | | | | All sources (t h$^{-1}$) | % oil and gas | % Non-oil and gas |
| **2021** | **7/10-8/11** | **5** | **99** | **87** | **50 (44 – 56)** | **95** | **2** |
| Permian-Delaware | 8/06-8/9 | 2 | 57 | 49 | 32 (27 – 37) | 100 | 0 |
| Permian-Midland | 7/10-8/9 | 3 | 39 | 35 | 15 (12 – 19) | 88 | 6 |
| Uinta | 8/11 | 1 | 3 | 3 | 1.9 (0.64 – 2.9) | 68 | 0 |
| **2022** | **11/1** | **1** | **4** | **4** | **5.7 (1.7 – 9.6)** | **95** | **5** |
| Anadarko | 11/1 | 1 | 4 | 4 | 5.7 (1.7 – 9.6) | 95 | 5 |
| **2023** | **6/1-10/13** | **40** | **323** | **268** | **363 (285 – 445)** | **80** | **18** |
| Anadarko | 10/7-10/8 | 2 | 12 | 11 | 13 (7.3 – 18) | 89 | 11 |
| Appalachian-Central | 7/31-9/5 | 5 | 91 | 55 | 52 (45 – 58) | 14 | 83 |
| Appalachian-North | 8/31 | 1 | 5 | 5 | 77 (4.5 – 149) | 100 | 0 |
| Arkoma Woodford-Caney | 6/23 | 1 | 2 | 2 | 5.7 (0 – 11) | 100 | 0 |
| Bakken | 6/4 | 1 | 1 | 1 | 0.28 (0 – 1.3) | 100 | 0 |
| Barnett | 6/24-10/6 | 2 | 8 | 8 | 12 (6.3 – 18) | 45 | 55 |
| Denver-Julesburg | 6/22-10/13 | 7 | 28 | 20 | 8.8 (6.1 – 11) | 29 | 71 |
| Eagle Ford | 6/1-6/28 | 3 | 18 | 18 | 21 (13 – 28) | 88 | 5 |
| Greater Green River | 8/28 | 1 | 6 | 6 | 8.3 (6 – 10) | 69 | 8 |
| Haynesville | 6/2-6/27 | 2 | 24 | 20 | 32 (22 – 43) | 100 | 0 |
| Permian-Delaware | 7/18-10/11 | 8 | 74 | 69 | 60 (43 – 78) | 100 | 0 |
| Permian-Midland | 6/10-10/12 | 4 | 41 | 40 | 54 (42 – 65) | 100 | 0 |
| Powder River | 9/27 | 1 | 3 | 3 | 8.6 (0 – 24.1) | 19 | 81 |
| San Juan | 8/22 | 1 | 4 | 4 | 6.1 (0.90 – 11.5) | 100 | 0 |
| Uinta | 7/16-8/26 | 2 | 6 | 6 | 1.9 (0.64 – 3.3) | 100 | 0 |

sources in the Denver-Julesburg had some of the lowest median emission rate (0.83 t h$^{-1}$, Figure 3A), driven by solid waste disposal facilities (landfills) and CAFOs.

Nationally, individual point source methane emission rates ranged from ~200 kg h$^{-1}$ to 70 t h$^{-1}$, highlighting the wide range in point source methane emission rates across basins and across sectors (Fig. 2, Fig. 3). Of the 426 sources observed in 2021-2023, 233 point source facilities (~55% of all point sources) emitted at mean emission rates >1 t h$^{-1}$ facility$^{-1}$, accounting for a cumulative total of ~84% of all detected point source emissions. In addition, only four methane "ultra" emitters (Lauvaux et al., 2022), with individual emission rates > 10 t h$^{-1}$, were responsible for ~20% of all point source methane emissions (see red curves in Fig. 2B, Fig. 2C), underscoring the skewed characteristics of point source emission distributions, as has been discussed elsewhere (Brandt et al., 2016). Overall, the confluence of observed emission ranges suggests individual large point source emission magnitude is shared across all facility types despite differing equipment and operational processes occurring on these sites. Delineation of specific categories within these types, e.g., transmission vs. gathering compressor stations or high vs low production wells, and/or the application of higher sensitivity instruments was outside the scope of this work, but is necessary to reveal possible emission rate stratification across facility sub-types.







**Figure 2. (a).** Box and whisker plot emission source distribution by facility type from 2021-2023. Points represent each emission source while numbers in bold below box plots indicate total sample size per category. The boxes represent the 25th and 75th percentiles, while the whiskers extend to 1.5x the interquartile range. Box plots are omitted for categories with a sample size of <6 total. **(b).** Cumulative emission rate distribution of all sources by facility types, ranked in ascending order of emission rates (e.g., the red line for all point sources show that facilities emitting <10,000 kg h⁻¹ contribute 80% of the total point source methane emissions quantified herein). Individual points represent each emission source. Facility types with a sample size of <6 total are omitted.





**Figure 3. (a)**. Box and whisker plot of emission source distribution by region from 2021-2023. Points represent each emission source while numbers in bold below box plots indicate total sample size per category. The boxes represent the 25th and 75th percentiles, while the whiskers extend to 1.5x the interquartile range. Box plots are omitted for categories with a sample size of <6 total. Box plots are omitted for categories with a sample size of <6 total. **(b)**. Cumulative emission rate distribution of all sources by region. Points represent each emission source, ranked in ascending order of emission rates (e.g., the red line for all point sources show that facilities emitting <10,000 kg h$^{-1}$ contribute 80% of the total point source methane emissions quantified herein). Individual points represent each emission source. Regions with a sample size of <6 total are omitted.



## 3.2 Ultra-emission events and basin-level point source emission characteristics

All named facility types, except waste sector facilities and CAFOs, had at least one observed emission source approximately
≥ 10 t h$^{-1}$ (Figure 2A), the range of what has been referred to as ultra-emitters (Lauvaux et al., 2022). In the Permian, we quantified a 48 ± 24 t h$^{-1}$ pipeline leak in the Delaware subbasin on 19 July 2023. According to air emissions event reporting from the New Mexico Oil Conservation Division (OCD Permitting Incident Details), our detection was a part of controlled blowdown of a gas gathering system in response to a detected leak. The gas gatherer reported total natural gas vented of 6000 Mcf with a methane fraction of 75% over a duration of two hours, from which we estimate a reported methane emission rate
of approximately 43 t h$^{-1}$, which is in reasonable agreement with our flyover quantification within uncertainty bounds.

In the northern region of the Appalachian, we detected a 69 ± 36 t h$^{-1}$ from a natural gas compressor station. As a result of this detection alone and only one overflight in the region, Appalachian-north had the highest observed basin-level point source flux (Figure 3A). These two detections alone notably skew the cumulative emissions distribution for pipelines and natural gas
compressor stations towards the fat-tail of the distribution (Figure 3B).

Even given multiple overflights, large but intermittent events can have an outsized impact on our understanding of a region's total emissions. Despite flying the core region of the Denver-Julesburg six times, one detection from a processing plant (8.0 ± 2.4 t h$^{-1}$) made up over a third (34%) of the basin's aggregated point source emissions from oil and gas sources (Figure 4).
Recurrence of site-level emissions was seen only from non-oil and gas facilities, suggesting that across the six overlapping flights all detected oil and gas facility level emissions were intermittent in the Denver-Julesburg basin in Colorado. The state of Colorado is often at the forefront for establishing oil and gas equipment standards and incorporating empirical observations into operator activity. The lack of recurrent emissions from oil and gas facilities could possibly be due to this regulatory environment, resulting in operators addressing leaks when detected.
Despite our survey's focus on oil and gas regions, MethaneAIR quantified a large portion of non-oil and gas sector emissions in many basins. In the Denver-Julesburg, over half of average total point source emissions above 550 kg h$^{-1}$ came from waste facilities (42%) and CAFOs (10%) (Figure 4B). In our single flyover of the Barnett, over half (54%) of point source emissions came from a plume at a power plant (6.8 ± 2.8 t h$^{-1}$). Coal sources make up the majority of emissions in the Powder River (81%), the largest source of coal in the U.S. (Luppens et al., 2015), and Appalachian-central which also had the largest
magnitude of non-oil and gas emissions (mean 43 t h$^{-1}$, 37 – 49 t h$^{-1}$ 95% CI) across the entire study. Over half of the coal facilities in the Appalachian were detected across multiple flights, with average persistence (detections / times overflown) being 0.54—the highest in the study. Waste facilities in the Appalachian-central also contributed a significant portion of emission sources above 550 kg h$^{-1}$ (12%). Overall, non-oil and gas emissions from multiple sectors make up significant portions of large point source emissions in these mixed basins, underscoring the importance of attribution or apportionment of
top-down data in these regions.





**(a)**

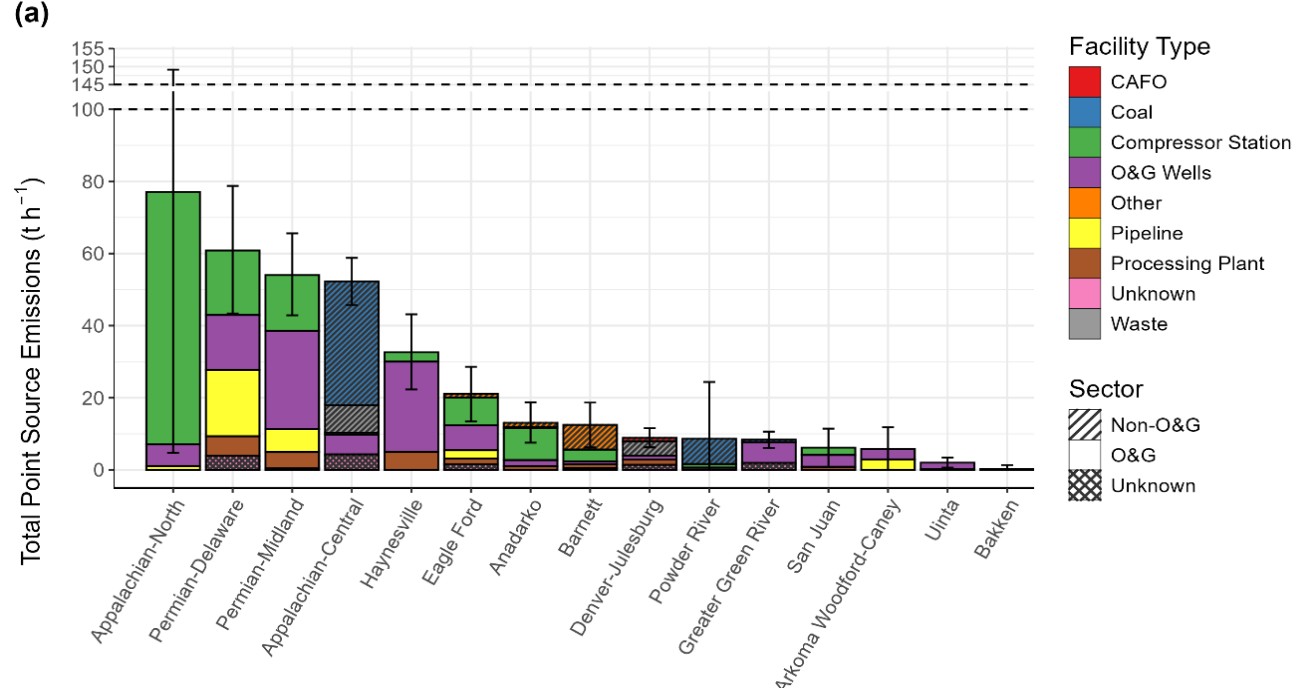

**(b)**

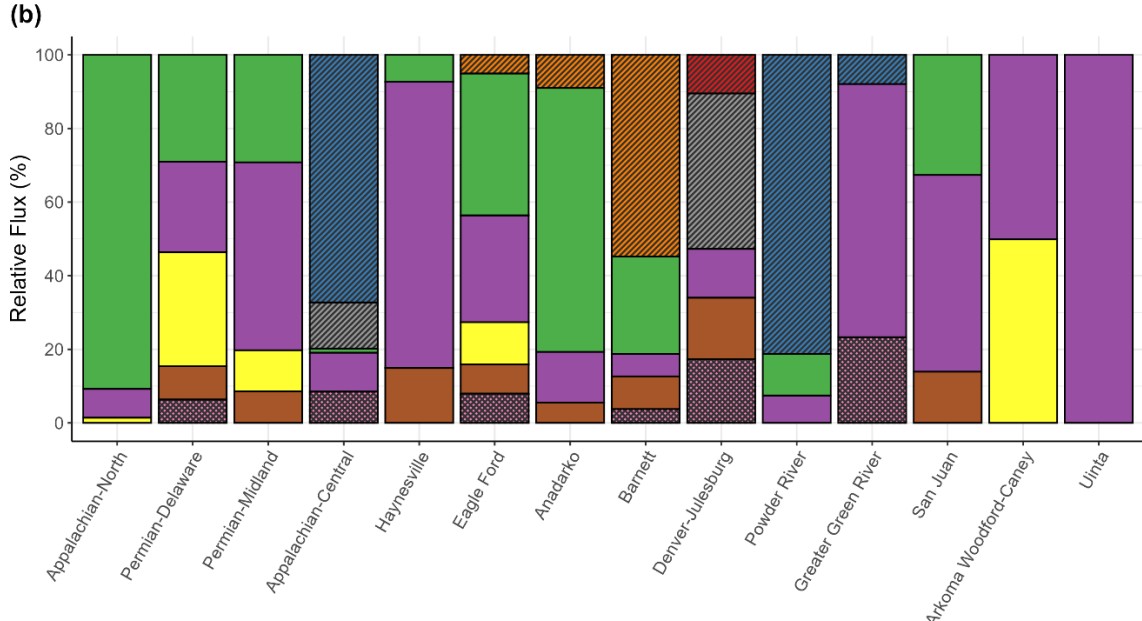

**Figure 4 (a).** Average basin-level emissions totals for all high-emitting methane point sources detected by MethaneAIR in 2023. Colors indicate specific facility types, while texture indicates non-oil and gas sectors. Error bars represent the 95%





confidence interval of the basin-level total emissions for all high-emitting methane point sources. Note the axis break between
100 t h⁻¹ and 145 t h⁻¹. **(b).** Normalized average basin-level emissions totals for high-emitting point sources above 550 kg h⁻¹
detected by MethaneAIR in 2023. For each region, the relative flux represents the ratio of each facility type's total methane
emissions as a fraction of the total emissions from all sources or facility types in the region.

285

Outside of the prior mentioned four regions, oil and gas sector sources make up the majority of emissions in all other studied
regions. Of the regions flown multiple times within the 2023 campaign, the highest magnitude of oil and gas sector point
source emissions was from the Delaware Permian (mean 59 t h⁻¹, 95% CI: 41–77 t h⁻¹) (Figure 4A). The Anadarko, despite
contributing approximately 6% to the nation's overall oil and gas emissions (Shen et al., 2022), has been largely unexplored
290    in multi-basin methane point source studies to date (Cusworth et al., 2022; Sherwin et al., 2024). Compressor station emissions
were the largest contributor to observed point source emissions in the Anadarko for both 2023 (mean: 8.8 t h⁻¹, 95% CI: 2.7–
14 t h⁻¹) (Figure 4) and MethaneAIR's research flight in 2022 (mean: 3.7 t h⁻¹, 95% CI: 0.71–6.3 t h⁻¹).

In the Haynesville and Eagle Ford, high-emitting methane point sources were concentrated in certain portions of the basin. In
the Haynesville, sources were more frequent in the Louisiana portion of the basin, while in the Eagle Ford, point sources were
focused in the southwestern-most area. Temporary events, such as a shutdown of midstream infrastructure, can lead to
widespread emissions for a short period. However, in the Haynesville, we sampled a large overlapping region 25 days apart
and saw a high frequency of point sources on both days, suggesting that our observations are not likely due to some kind of
short-term high emission events. Results from the Eagle Ford are based on a single overflight, and prior research in the region
has shown that aggregate emissions from large sources varied from two to three times on different days (Lavoie et al., 2017).
Even though MethaneAIR covered a much larger total area of the Eagle Ford than Lavoie et al., repeated scans are likely
necessary to more comprehensively characterize total and sectoral contribution of point sources.

Similarly, our results in the San Juan, Arkoma, Greater Green River, Powder River, Barnett, and Bakken are derived from one
comprehensive overflight and resulted in relatively few detections. Building a representative sample of total emissions and
sectoral contributions will likely require several overflights, and thus these results represent a baseline for future studies on
point source methane emissions in these basins.

The top six basins according to oil and gas point source methane emissions total—Permian, Appalachian, Haynesville, Eagle
Ford, Anadarko, Barnett (Figure 3) —are also the top six basins for basin level total oil and gas methane emission results, as
estimated from satellite-based inversions using TROPOMI (Shen et al., 2022) and GOSAT (Lu et al., 2023), as well as
measurement-based emissions inventory (Omara et al., 2024). However, across all three studies there are uncertainties in the
ordering of these basins by total oil and gas emissions, except for the Permian which is consistently observed as the highest
emitting basin in the U.S. Future investigations with simultaneous quantification of both the total area emissions and high-



emitting methane point sources are needed to better characterize the relative contribution of point sources to the overall emissions, which are expected to vary across basins (Williams et al., 2024).

## 3.3 High-emitting point source total and sectoral contribution and total variability over time

Our estimate of 113 t h$^{-1}$ (95% confidence interval: 86–144 t h$^{-1}$ ) of methane emissions from Permian point sources from MethaneAIR in 2023 initially appears significantly higher than the estimate reported in Cusworth et. al 2022 for the year 2021

(67.7 ± 19 t h$^{-1}$, summer 2021; 74.1 ± 27 t h$^{-1}$, fall 2021) despite the relatively lower detection threshold of the AVIRIS-NG instrument used in Cusworth. However, the presented values from Cusworth et al. 2022 cover a different spatial extent and use a different definition of persistence, with an additional step of Monte Carlo simulations which is not applied here. If we limit our comparison to only the overlapping core areas of the Delaware and Midland observed in both studies, apply the same calculation methods, and use a minimum threshold of 550 kg h$^{-1}$ to align sensor sensitivities, MethaneAIR's 2023 95%

confidence interval for average total point source emissions estimate in the Delaware (12–18 t h$^{-1}$) is consistent within statistical uncertainty with Carbon Mapper's estimate in summer 2021 (17–20 t h$^{-1}$) and MethaneAIR's research flights in 2021 (14–21 t h$^{-1}$) in the same region. Overall, 2023 average total point source emissions in both the Permian Delaware (12–18 t h$^{-1}$) and Midland (3.6–12 t h$^{-1}$) show a decreasing trend from measured highs in 2019 as measured by CarbonMapper (Midland: 39–45 t h$^{-1}$, Delaware: 50–54 t h$^{-1}$) (Figure S6.1).


Outside of the Permian, Uinta point source emissions above 550 kg h$^{-1}$ are consistent and time-invariant across Carbon Mapper's 2020 campaign (0.52–2.2 t h$^{-1}$), MethaneAIR's 2021 research flight (.007–2.2 t h$^{-1}$), and MethaneAIR's 2023 flights (0.0–1.7 t h$^{-1}$). Average total point source emissions in Appalachian-Central from overlapping regions show a decrease from Carbon Mapper's observations in 2021 (58.5-71.5 t h$^{-1}$) to MethaneAIR's in 2023 (22.7–32.1 t h$^{-1}$). MethaneAIR's point source

emissions totals in the Denver-Julesburg for 2023 (2.3–4.5 t h$^{-1}$) is between estimates from Carbon Mapper in summer (1.5–2.9 t h$^{-1}$) and fall (4.1–5.3 t h$^{-1}$), underscoring the variable nature of point source estimates in specific basins due to the underlying intermittency of sources even in shorter timeframes. While these comparisons illustrate changes in point source emissions over time, emissions trends in the overlapping regions of these studies may not necessarily represent emissions trends of the entire basin or cumulative emissions from all emission rates and sources.


When comparing the relative sectoral contributions of point source contributions over time, we see a broad level of consistency from MethaneAIR 2023 to results in Cusworth et al. 2022 from prior years (Figure S5.1). The same facility types are represented each year with relatively small fluctuations over time when considering uncertainty. Our results suggest an increase in waste sector point source emissions in Denver-Julesburg basin for 2023 relative to the 2021 Carbon Mapper observations.

Given that many large point sources represent abnormal process conditions and are highly intermittent, we expect inherent variability in a region unless a highly persistent facility type is abundant. This expectation is confirmed when looking at results in the Appalachian, where 60% of high-emitting point sources emissions are from coal facilities in both 2021 and 2023.





### 3.4 Implications for policy and future scientific work

This study investigates the relative contribution of methane emissions from various facility types using high-emitting point
source detections from MethaneAIR. However, contribution by a facility type can change depending on the observed portion
of the emissions distribution. For example, low-producing wells are estimated to make up half of all production-related
emissions in the U.S., primarily from sources emitting below 5 kg h$^{-1}$ (Omara et al. 2022). Given the relatively high detection
threshold of point sources for MethaneAIR (~200 kg h$^{-1}$; Chulakadabba et al., 2023), certain sub-facility types are not
characterized by this study. Extensive sampling with instruments of lower detection limits (Johnson et al., 2023) as well as
additional processing methods of existing data (Guanter et al. 2024) offer an avenue for comprehensive investigation across a
larger distribution of emission rates. Under recently finalized regulations, U.S. EPA will begin requiring the reporting of large
methane releases under the Super-Emitter Response Program (SERP), including emissions with an instantaneous emission rate
of at least 100 kg h$^{-1}$ (EPA, 2024b). Therefore, all plumes within this study would be reported if collected by a SERP reporter.
While our results indicate MethaneAIR's sensitivity does not comprehensively extend to the lower limit of SERP's threshold,
the attribution analysis presented here begins to illustrate what we can begin to expect from the program.

As a precursor mission to MethaneSAT, MethaneAIR and its point source observations offer a baseline for the satellite to
expand with its observational capacity, while also improving what we can interpret from the satellite data. National campaigns
with an airborne platform like MethaneAIR, even with a relatively large observational swathe, are limited in the number of
repeat overflights to a given region. Representative sampling to characterize overall emissions variability can require multiple
overpasses (Chen et al., 2024; Lavoie et al., 2017), and characterizing the sectoral contributions magnifies this sampling need
depending on the intermittency of the underlying facility types. The revisit frequency and ability to observe multiple targets in
a single day using satellite platforms, such as MethaneSAT (www.methanesat.org) and the upcoming CarbonMapper satellites
(Carbon Mapper, 2024), will provide a solution to this sampling need. Facility-level and sectoral source attribution is possible
with satellite-observed point sources, depending in part on each instrument's spatial resolution specifications for point source
quantification. More research and application of source-apportionment methods (Carranza et al., 2022; Fiehn et al., 2023) is
needed, particularly for low-emitting sources that will appear as diffuse area emissions in top-down inversions.

### 4 Conclusion

Herein we presented results on high-emitting methane point sources from the airborne campaigns covering over 80% of U.S.
onshore oil and gas production based on MethaneAIR measurements. Using detailed facility-level attributions, we investigated
the sectoral contribution of point sources, including in regions previously unexplored in the methane point source literature.
Our results show a tremendous amount of variation in facility-type contribution to point source methane emissions, even
among similar basins in terms of oil and gas play characteristics. While most oil and gas producing regions are dominated by

oil and gas methane point sources, we find that non-oil and gas sources can make up a significant proportion of high-emitting totals in several basins. Detections from extremely large and intermittent sources can disproportionately affect basin and facility-level insights, underscoring the importance of collecting a large sample size to more comprehensively assess the frequency of these events. Due to the underlying intermittency of high-emitting point sources and variation across facility types, additional sampling is needed for many regions where observations were limited to a few comprehensive flights.

Nonetheless, our results offer a comprehensive baseline for future investigations on the sectoral contribution of emissions, which will be greatly improved by the observational capabilities of upcoming satellite-based data.

## Data availability

MethaneAIR point source data can be accessed online from the Earth Engine Data Catalog at

https://developers.google.com/earth-engine/datasets/tags/methaneair

## Code availability

Python 3.12 and R code used for data analyses and visualizations can be obtained from the corresponding authors upon reasonable request.


## Author contributions

Conceptualization and design of this study was led by RG and SW. Processing and development of the MethaneAIR observations was conducted by MS, CCM, SR, EM, AC, JB, ZZ, and SW. JDW performed source attribution and analysis and visualizations of high-emitting methane point sources, with contributions from JPW, MO, AH, KM, and LG. JDW wrote the

manuscript, with contributions from all authors.

## Competing interests

The corresponding authors declare that none of the authors has any competing interests.

## Acknowledgements

We acknowledge funding support from the Bezos Earth Fund.

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
