# Peer review of "Sectoral contributions of high-emitting methane point sources from major U.S. onshore oil and gas producing basins using airborne measurements from MethaneAIR"

_EGUsphere, 2024_

## Author Comment (AC1)

We thank the reviewers for their insightful comments. Responses are in blue, while the original comments are in black italics.

RC1

*This study describes the results from MethaneAir airborne campaigns across multiple basins in the U.S., with emphasis purely on attributed point sources above the MethaneAir detection limit. The study clearly presents the algorithms, the attribution, the results, and performs inter comparisons with other studies. I have a few comments that need to be addressed before I can recommend for publication.*

1. *Although this study represents (to the authors' knowledge) the largest airborne methane survey performed over onshore oil&gas in the U.S., it wasn't entirely clear what new learning was gained from such effort.*

- *(1a) One possibility is in their section on repeat sampling in Haynesville, where they show that point source frequency is possibly stable.*

Both reviewers commented to some degree that the scientific question and corresponding central findings of this study are not readily apparent in its current form. The scientific question of this study was articulated starting from line 68 as, "we investigate how high-emitting methane point sources are distributed across facility types, and if any patterns of emissions emerge across different regions with varying mix of oil and gas and non-oil and gas methane sources".

Per the reviewers' comments, this broad question does not sufficiently highlight presented results and enable a reader to evaluate whether the study was successful in answering its scientific question(s).

To better address this shortcoming, we have restated the scientific questions in in the final paragraph of the introduction (lines 70-74) as follows…

  *(i)    What is the estimated distribution and contribution of high-emitting methane point emissions for basins in the U.S., and have these basins been explored in the prior literature?*
  *(ii)   For basins previously explored in the literature, is there evidence that high-emitting methane point source emissions have changed over time?  How does the relative sectoral contribution of high-emitting methane point sources vary basin to basin?*

We believe this framing better matches the analysis, results, and discussion, and that there are clear connections between the questions asked and figures and the discussion presented.

New learnings from these investigations includes methane point source quantification and sectoral attribution from multiple basins either currently not present in the methane point source literature—Anadarko, Haynesville, Arkoma, Bakken, Powder River, Greater Green River—and multiple basins that have been studied in the past but lack either recent or detailed analyses of point sources—Barnett, Eagle Ford. Other studies, namely Cusworth et al. 2022 and Sherwin et al. 2024, present results on point sources from multiple basins—Appalachian, Permian, Denver-Julesburg, Uinta, Barnett, San Juan—for years prior to 2023. This study builds upon prior multibasin work through the temporal comparison of 2021-2023 MethaneAIR data to overlapping sampled regions to Cusworth et al. 2022. We consider our findings on the temporal trends of point emission sources through 2021-2023—decreasing overall emissions in the Permian and Appalachian-Central, level emissions in the Uinta, and consistent overall magnitude but fluctuations in contributions from individual facility types in the Denver-Julesburg—to be novel contributions to the scientific literature on high-emitting methane point source characteristics.

As noted by the reviewers, the study highlights several basin-specific insights—such as the consistent frequency of point sources in the Haynesville, or the lack of persistence among oil and gas sources in the Denver-Julesburg. However, as the overall scope of this study is on U.S. wide methane point sources, pertaining to surveyed oil and gas regions responsible for over ~80% of onshore production, our central findings are on point source emissions total per basin and the proportional contribution from oil and gas versus non-oil and gas sources (Table 1).

We have made several edits throughout the manuscript to accordingly clarify the research question, novel contributions, and central findings following these ideas.

- (1b) *Another possibility is that non-oil&gas point sources make up an equal fraction of emissions as oil&gas point sources. That is very interesting, but the authors need to address detection limit issues if that's the central thrust of the paper. For example, certainly there are a lot of oil&gas point sources not detected by MethaneAir that were there (hence the need to filter results for comparison with Cusworth et al., 2022). For other sectors, if the general distribution of emissions is higher than the MethaneAir detection limit (e.g., underground coal vents), then those would be readily detected while some oil&gas sources wouldn't be detected, hence over-inflating the non-OG contribution to point sources. In any event, a presentation of this dataset itself is interesting, but some clarity on the general scientific questions this study is answering would be helpful to contextualize the results.*

The reviewer is correct in noting that non-oil and gas point sources make up the majority of observed total point source emissions, but only in certain basins (Appalachian-Central, Denver-Julesburg, Powder River, Barnett) and only when using this study's emissions threshold for comparison.

We respectfully disagree with the suggestion that the contribution from non-oil and gas sources, particularly coal in the Appalachian-Central, are overinflated in this study. In our analyses, we account for present but unidentified sources by using the emissions detection threshold for comparison, assessed based on the measurement data herein, which in turn ensures that our assessment of the relative sectoral contributions of high-emitting methane point sources are robust for sources emitting above this threshold. We believe we are very forthcoming about what portion of the emissions distribution curve is being analyzed, while highlighting that the relative contribution of facility types to total emissions will fluctuate across the broader emissions distribution curve. This contextualization relative to the method's emission rate threshold for comparison does not detract from our study, as nearly all aircraft or satellite imaging platforms focused on point source characterization share this caveat or limitation

*2. Line 85. A more recent MethaneAir paper cites 33-38 ppb for MethaneAir precision in the Permian Basin and Arizona. Why is there a discrepancy? Is 17-20 ppb consistent with what you calculated in this study? It would be nice to know the background standard deviation in each of the regions you surveyed as it contextualized the distributions of detections in this manuscript.*

We thank the reviewer for noting the differences between the two studies. Chan Miller et al., 2024 33-38 ppb precision is based on L2 retrievals at a 20m x 20m image. Meanwhile, Chulakadabba et al., 2023 reports 17-20 ppb standard deviation of gaussian denoised L3 images at 10m x 10m pixels for a separate set of flights. While there are inherent differences between what is estimated Chan Miller et al., 2024 and Chulakadabba et al., 2023, we do expect variation in the retrieval precision from flight to flight. For the flights included in this study, we found the pixel precision for L3 images to be in the range of 25-35 ppb depending on the flight conditions and light levels for the scene. This result has been added to section 3.1.

*3. Lines 108-114. The DI and clumping technique you describe for plume detection uses many hard-coded values (e.g., 600-m pixel squares, 12 pixel clumps, etc.). Is this approach exactly the same as what was summarized in El Abbadi et al., 2024? In other words, how sensitive is the algorithm to these assumptions?*

Those hard-coded numbers were all tested over a range of values, and the chosen values represent what found the most true plumes with the least number of false positives in our testing. The algorithm is somewhat sensitive to the values chosen in that other values may find slightly more true plumes, but with a lot of false positives, or may miss more plumes. The clumping technique for finding plumes was not used in El Abbadi et al. 2024, because in that case it was a controlled release experiment, and the location of the plumes was known. This method for finding plumes without prior knowledge of source locations has not previously been published.

*4. Related to comment #2 - by selecting 600m windows, you are making a prior assumption on plume size which then could bias your detection algorithm. Put another way, if you assumed 100-m windows (i.e., smaller plume sizes), would this result in more plume detections? Have you tested this?*

Yes, we have tested windows of various sizes, and 600m windows seem to best match the scales of plumes we are able to detect and quantify with this instrument. The value was chosen as our testing showed that it produced the most true plumes with the least false positives.

*5. Line 261-264. This is conjecture. I don't think that conclusion is defensible without citation or actual operational data.*

We agree and apologize for the original phrasing of these lines without contextualization or citation. The discussion point has been revised as follows to cite both the regulations and a recent study that uses process model simulations to explore their possible impacts on oil and gas emission reductions.

*"Prior research using process level models of oil and gas emissions that takes into account the changes in the regulatory environment suggests that production normalized gas loss rates have*

*decreased in response to regulatory requirements starting in 2014 (Riddick et al., 2024). The observed lack of recurrent emissions ≥150-550 kg/ h$^{-1}$ from oil and gas facilities could possibly be attributable to this regulatory environment, which has included an empirically based oil and gas methane intensity verification program that was adopted in 2023."*

*6. Line 353 and elsewhere. You cite the detection limit for MethaneAir to be ~200 kg/h repeatedly throughout the text, but then restrict most inter comparison to emissions above 550 kg/h. Other studies name this specifically with a probability of detection or an effective detection limit. You've shown in this study that effectively the detection limit for methane air is 550 kg/h, so it would be more accurate to update every mention of ~200 kg/h to "200-550 kg/h."*

We thank the reviewer for the suggestion and noting the needed clarity for emission rates mentioned.

The minimum detection limit of this methodology was defined as 150 kg/hr in section 2.2, which we determined based on both the estimated limit for identifying plumes in Chan Miller et al., 2024 and the limit for reliably quantifying plumes from Chulakadabba et al., 2023. Using simulated XCH4 data, Chan Miller et al., 2024 found that the median detection limit for identifying plumes based on a MethaneAIR research flight scene (RF04) was 121 kg/hr (interquartile range: 106-141 kg/hr). This limit for identifying plumes is consistent with the 200 kg/hr threshold proposed by Chulakadabba et al., 2023, which was based on the results that estimated emissions below 200 kg/hr were uncertain and can be overestimated relative to known emission rates in a controlled release experiment.

Mentions of the method detection limit being ~200 kg/hr do not clearly reflect treatment of the data and have been standardized to 150 kg/hr in the text.

In section 2.4 (Analyses of high-emitting methane point source emissions), we introduce a threshold for comparison that is based on the methodology used in Chen et al. 2024. Chen et al. refers to this minimum threshold as the "size limit for direct comparison", so we refer to it as the threshold for comparison and the range of 150-550 kg/hr as the partial detection range. Beyond this nomenclature, we've attempted to contextualize results and takeaways with specific mention of the emission rate class (e.g. "for point sources above 550 kg/hr…").

Mentions of the estimated threshold for comparison have been shifted from methods section 2.4 to results section 3.1. Additionally, we have added another figure to the supporting information (Figure S5.3) that clarifies the effective flux contribution per basin from sources in this partial detection range.

---

## Author Comment (AC2)

We thank the reviewer for their insightful comments. Responses are in blue, while the original comments are in black italics.

RC2:

*This manuscript describes the results from a large MethaneAIR campaign, an aerial methane measurement method, across several U.S. onshore production basins. The results from this paper focus on the distribution of high-emitting methane point sources across different types of facilities, basin-level trends in point source behavior, and understanding the relative contributions of methane point sources across different sectors. I believe this manuscript provides data that will advance our understanding of methane emissions and distributions, but would benefit from additional analysis. It is not immediately clear what the novel contribution of this work is, and additional analyses could be conducted to make this manuscript more compelling. Some suggestions for additional investigation would be:*

Both reviewers commented to some degree that the scientific question and corresponding central findings of this study are not readily apparent in its current form. The scientific question of this study was articulated starting from line 68 as, "we investigate how high-emitting methane point sources are distributed across facility types, and if any patterns of emissions emerge across different regions with varying mix of oil and gas and non-oil and gas methane sources".

Per the reviewers' comments, this broad question does not sufficiently highlight presented results and enable a reader to evaluate whether the study was successful in answering its scientific question(s).

To better address this shortcoming, we have restated the scientific questions in in the final paragraph of the introduction (lines 70-74) as follows…

*(i)     What is the estimated distribution and contribution of high-emitting methane point emissions for basins in the U.S., and have these basins been explored in the prior literature?*

*(ii)    For basins previously explored in the literature, is there evidence that high-emitting methane point source emissions have changed over time?  How does the relative sectoral contribution of high-emitting methane point sources vary basin to basin?*

We believe this framing better matches the analysis, results, and discussion, and that there are clear connections between the questions asked and figures and the discussion presented.

New learnings from these investigations includes methane point source data from multiple basins either currently not present in the methane point source literature—Anadarko, Haynesville, Arkoma, Bakken, Powder River, Greater Green River—and multiple basins that have been studied in the past but lack either recent or detailed analyses of point sources— Barnett, Eagle Ford. Other studies, namely Cusworth et al. 2022 and Sherwin et al. 2024, present results on point sources from multiple basins—Appalachian, Permian, Denver-Julesburg, Uinta, Barnett, San Juan—for years prior to 2023. This study builds upon prior multi-basin work through the temporal comparison of 2021-2023 MethaneAIR data to overlapping sampled regions to Cusworth et al. 2022. We consider our findings on the temporal trends of point emission sources through 2021-2023—decreasing overall emissions in the Permian and Appalachian-Central, level emissions in the Uinta, and consistent overall magnitude but

fluctuations in contributions from individual facility types in the Denver-Julesburg—to be novel contributions to the scientific literature on high-emitting methane point source characteristics.

As noted by the reviewers, the study highlights several basin-specific insights—such as the consistent frequency of point sources in the Haynesville, or the lack of persistence among oil and gas sources in the Denver-Julesburg. However, as the overall scope of this study is on U.S. wide methane point sources, pertaining to surveyed oil and gas regions responsible for over ~80% of onshore production, our central findings are on point source emissions total per basin and the proportional contribution from oil and gas versus non-oil and gas sources (Table 1). The prevalence of non-oil and gas point source emissions totals relative to the emission rate threshold for comparison, and how they make up most observed emissions in several basins is also a key finding that has not been explored previously at this scale using methane point source imagers.

We have made several edits throughout the manuscript to accordingly clarify the research question, novel contributions, and central findings following these ideas.

*- Further explore the contribution of emissions from other (non oil and gas) sectors. Are there other previous studies from these sectors that could be included for comparison?*

Other non-oil and gas sectors that are grouped together but not explicitly named in the figures including power plants, a biogas facility, and a fertilizer plants. Comparisons of the power plant and fertilizer plant observations to prior studies have been added to the discussion.

*"MethaneAIR observed emissions from several other facilities not commonly considered for their methane emissions, including power plants, a biogas storage facility, and a fertilizer plant. Plumes were detected at both a coal-fired power plant in the Eagle Ford and a natural gas-fired power plant in the Barnett. Emissions detected at the natural gas-fired power plant ($6.8 \pm 2.8$ t h$^{-1}$) exceed estimated methane emissions rate of uncombusted natural gas from typical stack operations in prior measurement based work (8-135 kg h$^{-1}$) (Hajny et al., 2019), suggesting MethaneAIR's observation was a result of upset conditions or a separate fugitive source."*

*"MethaneAIR also observed emissions from a fertilizer plant in the Anadarko that was previously sampled using mobile surveys in 2016 (Zhou et al., 2019). Emissions quantified by MethaneAIR in 2023 ($1.1 \pm 0.4$ t h$^{-1}$) greatly exceed the prior estimated fertilizer plant emission rate from mobile survey sampling across two days ($213 \pm 118$ kg h$^{-1}$)."*

*-Further exploration of the ultra-emitters (Line 213)*

Ultra-emitters (> 10 t h$^{-1}$), and their disproportionate effect on basin-level emissions magnitudes and relative contributions of emissions by facility types are discussed in the first three paragraphs of section 3.2 Ultra-emission events and basin-level point source emission characteristics. They are also discussed specifically in lines 215-216, noting that four ultra-emitters make up ~20% of all observed point source methane flux. We also go into depth about one detection, comparing its reported emission rate to a state agency to the emission rate quantified by MethaneAIR.

*-For regions with multiple overpasses, how do they day-to-day statistics compare? Earlier in the manuscript (Line 137), it is mentioned that with large enough sampling you could assume representative sampling across space and time (ergodic hypothesis). Do you find this to be true*

*in regions where there were multiple overpasses? Can anything be learned about minimum requirements for sampling, and the equivalency between spatial and temporal sampling?*

We thank the reviewer for the suggested questions. The line referenced above pertained to the applicability of random sampling persistence values through a Monte-Carlo approach for weighting emissions in cases where there were few overlaps, and not about the minimum sampling requirements for developing a representative estimate of basin-level point source emissions. Section 2.1 briefly discusses sampling requirements and how that was accounted for in the planning phase for the aircraft campaigns.

In light of these questions, we have added an additional supporting information section that explores the day-to-day variation in select subregions with the highest number of overflights to expand on how regional emissions variation and intermittency affects the number of samples needed for consistent or representative emissions estimates.

The below figures show total emission estimates for a large subregion of the Appalachian-Central flown on five days and another subregion in the Denver-Julesburg flown on six days.

For the Appalachian-Central subregion, daily and aggregate emission estimates all overlap with one another given their respective 95% confidence intervals. This result is in part due to the large uncertainty ranges on the day-to-day estimates. However, any combination of two days would produce a total flux estimate without uncertainty that is within the confidence interval of the five-day estimate. We interpret these results to mean that for the emissions observed by MethaneAIR using these processing methods, one to two sampled days would produce a representative estimate of point source emissions for this subregion. Whether or not this interpretation applies to the entire basin would depend on changes in the observed facility composition and source intermittency going from the subregion to basin scale.

Conversely, results in the subregion of the Denver-Julesburg show a much larger degree of day-to-day variation. Observations from 10-09-2023 and 10-13-2023 are comparatively lower than the estimate from 06-22-2023, which had a single processing plant plume (~8 t/h) contributing most emissions on that day. Additionally, we cannot rule out the possibility of changes due to seasonal variation in the approximately three-month time period between the first and last flights. The range in the flux estimation without uncertainty using any combination of five out of six sampling days produces a larger range than that of the 95% confidence interval of the estimate using all six days. Assuming the observed variation is strictly due to the intermittency of methane point sources and not seasonality, we interpret this result to signal that several sampling days, possibly five or more, are required to estimate total point source emissions in this emission rate range for this subregion.

Our results in the Appalachian and Denver-Julesburg basins, although from different basins with very different facility type composition, suggest that required temporal sampling will vary by basin and possibly by subregion. This finding is consistent with results from Chen et al. 2024, which indicated that emissions characteristics in the production core vs periphery of the New Mexico Permian were significantly different, and that one comprehensive overflight may not be enough to estimate emissions depending on the desired accuracy. While only one daily estimate, 10-09-2023, produced an uncertainty range that did not overlap with the uncertainty range using all observed days, this result is primarily driven by large uncertainty ranges of daily and aggregate emissions estimates and not necessarily justification for the application of an ergodic hypothesis based on one comprehensive overflight.

[Figure]

Figure S6.1 Daily variation of point source emission from a subregion of the Appalachian-Central. All represents the total point source emission estimate using all days with persistence weighting.

[Figure]

Figure S6.2 Daily variation of point source emission from a subregion of the Denver-Julesburg. All represents the total point source emission estimate using all days with persistence weighting.

*In addition to the issue of scientific novelty, I also have specific comments related to the technical aspects of the work that I suggest be addressed before publication:*

*- Line 47: "Methane point sources above 10kg/hr can make up 13-59% of total regional flux" Please clarify the meaning of this statement. Are these point sources from all sources or just oil and gas sources? Is this just for oil and gas producing regions? What is the remaining 87-41% - is it from diffuse sources or source below 10kg/hr?*

We thank the reviewer for the clarifying questions. The original statement, "Research by Cusworth et al. (2022) comparing point source quantifications to overlapping satellite-based inversions showed that methane point sources above 10 kg h$^{-1}$ can make up 13-59% of total regional flux in the U.S" was a reiteration of the range of "Contribution of point sources to area flux (%)" values from Table 1 in Cusworth et al. 2022 alongside their stated upper end of their minimum detection limit in the study. These totals come from all sectors, including oil and gas, waste, coal, and agricultural emissions (CAFOs). In the study, the difference in flux between the satellite inversion results and the point source totals is the remaining 87-41% of flux, depending on the basin and individual flight campaign. We could interpret the 87-41% remainder as dispersed area emissions from sources both below 10 kg/hr and unidentified sources in their partial detection range. We have revised the sentence as follows:

*"Research by Cusworth et al. (2022) comparing point source quantification to overlapping satellite-based inversions showed that observed methane point sources--above a minimum*

*detection limit of 10 kg/hr--from all sectors can contribute up to 13-59% of total regional flux in certain basins."*

*- The detection threshold is defined as 550 kg/hr, but data was included in the analyses and figures for emissions of ~200 kg/hr. Emissions below 150kg/hr were discarded. Please elaborate on this treatment of the data including the justification for excluding emissions below 150 kg/hr, while including emissions below 550 kg/hr. Is there detection testing that supports this?*

To briefly summarize the thresholds as presented in the original manuscript.

Lines 120-121: "Plumes with a flux less than 150 kg h$^{-1}$ were discarded as being below the detection threshold of the methodology. In addition, we manually reviewed all identified plumes and discarded plumes with known artifacts."

Using simulated XCH4 data, Chan Miller et al., 2024 found that the median detection limit for identifying plumes based on a MethaneAIR research flight scene (RF04) was 121 kg/hr (interquartile range: 106-141 kg/hr). This limit for identifying plumes is consistent with the 200 kg/hr threshold proposed by Chulakadabba et al., 2023, which was based on the results that estimated emissions below 200 kg/hr were uncertain and can be overestimated relative to known emission rates in a controlled release experiment. Based on these limits for identifying and quantifying plumes, we define the detection limit for this methodology at 150 kg/hr and discard quantified emissions below said threshold.

Mentions of the detection limit being ~200 kg/hr do not clearly reflect treatment of the data and have been standardized to 150 kg/hr in the text.

The emission rate threshold for comparison (Lines 159-162), defined by the peak observational frequency (Supporting Information S5) using a similar approach as Kunkel et al., 2022 and Chen et al., 2024, was approximately 550 kg/hr. We use this threshold as an approximation for a high probability of detection across all observing conditions. A standalone work is in development for quantitatively exploring probability of detection, as was done in Conrad et al., 2023 and Ayasse et al., 2024.

Emissions in the range of 150-550 kg/hr therefore represent the partial detection range for this methodology. To clarify the effect the partial detection range has on our estimates of total emissions, we have added some language in the main text and the below figure in the supporting information which is an alternate form of Figure 4a that shows the effective contribution from plumes above and below 550 kg/hr to basin level flux. Overall, the partial detection range has a minimal effect on estimated point source emissions, contributing only ~2.4% of the total flux estimate.

[Figure]

Figure S5.3 Average basin-level emissions totals for all high-emitting methane point sources detected by MethaneAIR in 2023. Colors indicate flux contribution from sources within the partial detection range (blue) and above the threshold for comparison (red). Numbers below the stacked bar chart represent total plume sample size. Error bars represent the 95% confidence interval of the basin-level total emissions for all high-emitting methane point sources. Note the axis break between 100 t h$^{-1}$ and 145 t h$^{-1}$.

Additionally, language has been added throughout to ensure that resulting conclusions are recognized alongside the detection threshold for comparison, or in other words the portion of the emissions distribution curve that was observed.

Also in section 3.1, we have revised the previous approximation of detected emission rate range to actual emission rate range (160 kg/hr to 70 t/h)

*I would also consider the probability of detection as well, rather than assuming a binary detect/non-detect. What is the probability of detecting a 550kg/hr source vs a 200 kg/hr source? If a site that was previously detected at 200kg/hr is not detected again, how can you discern whether this is because of a low probability of detection or because the source stopped emitting?*

The assessment of a discreet probability of detection using a continuous function given source rate, wind speed, and flight altitude derived from controlled-release observations as done in Conrad et al. 2023, was outside the scope of this study.. However, research on estimating

quantitative probability of detections for MethaneAIR using similar frameworks as in Conrad et al. 2023 and Ayasse et al. 2024 was presented at AGU 2024 (https://agu.confex.com/agu/agu24/meetingapp.cgi/Paper/1721700) , and is currently in development for a standalone study. Preliminary results indicate that 550 kg/hr represents a high (>90%) POD given the observing conditions of the 2023 MethaneAIR flights.

*- Line 260: Related to above comment: Could the intermittency of detection be due also to probability of detection? Is it fair to conclude that just because the sources were not detected, they were not there? The manuscript states that "Recurrence of site-level emissions was seen only from non-oil and gas facilities" - was there 0 recurrence at oil and gas facilities?*

The reviewer is correct in pointing out the possibility of emitting but undetected sources within the partial detection range and below the minimum detection limit. Lines 260-261 have been edited accordingly and additional language added clarifying this distinction where persistence and individual facility level emissions are discussed.

Line 260-261

"Recurrence of site-level emissions was seen only from non-oil and gas facilities, suggesting that across the six overlapping flights all detected oil and gas facility level emissions were intermittent in the Denver-Julesburg basin in Colorado."

Revised to

*"Recurrence of site-level emissions was seen only from non-oil and gas facilities. While it is possible that there were undetected but present emissions below this study's effective detection limit, our results indicate that emissions from oil and gas facilities were all single occurrences in this size class for six overflights covering the core part of the basin."*

*- Line 133: Please explain why the emission rate was divided by the number of overpasses. It might be more helpful to report the average emission rate for detected emissions and the number of non-detects separately. The number of "0" emission measurements, especially when there were previous emissions there, can be just as important as the measured emission rates. This also relates to the first comment of understanding whether non-detects/"0" measurements are a function of probability of detection or source intermittency.*

Emission rate is divided by the number of overpasses when calculating a basin's total point source emissions in order to normalize the contribution from subregions with more overflights relative to those with fewer. The interest of this study is to characterize the relative contribution of high-emitting point sources by facility type and sector across major U.S. oil and gas basins. We minimize the effect of true absence vs present but not detected emissions for basin-level estimates by limiting portions of the analysis by the threshold for comparison. Comparisons on the relative contribution of facility types (Figure 4b) and temporal comparisons (Figure 5) are limited by the threshold for comparison for these observations (550 kg/hr). Total emissions magnitudes by basin (Table 1 and Figure 4a) are not filtered by the threshold for comparison, but the effect is minimal (see prior comment, 2.4% of total weighted flux comes from detections below 550 kg/hr) and is broken down in the additional supplemental section.

Note, plumes are filtered prior to weighting by overflights (e.g. a plume detected at 1 t/h flown over five times contributes only 200 kg/hr of flux to basin level totals in Table 1 and Figure 4a & b. While the effective contribution of the plume is below 550 kg/hr, it is still present in Figures 4b and ).

While contextualizing the analysis and results relative to the threshold for comparison limits the impact from sources with a low probability of detection, intermittency and flux variation at the regional scale still influences our results on basin-level estimates. Per a prior reviewer comment, this is explored in the newly added supporting information section no day-to-day variability.

*- Table 1: Does the % oil and gas relate to the count of detections or the emissions contributions* Aggregated total flux.

% oil and gas relates to the emissions contributions to aggregated total flux. Headers have now been modified to "% oil and gas flux" and "% Non-oil and gas flux".

*- Line 208: Is there a reason why the DJ basin has the highest frac of non-oil and gas in Table 1, but the lowest emission rates?*

The low median emission rate for the Denver-Julesburg (Figure 3a) and high fraction of non-O&G gas emissions (Table 1) is driven by the fact that most of the plumes are coming from waste and CAFO facilities, which as shown in Figure 2a have the lowest median emission rates of all explicity named facility types. Proportion of total point source flux from non-oil and gas sources is called out in Table 1, while individual plume emission rates and oil and gas versus non-oil and gas attribution are visualized in Figure 3a.

*- Figures 2 and 3: Blue/green colors are hard to distinguish from one another*

Thank you for the suggestion. The color map for figures 2 and 3 have been updated to match figure 4 and 5 for consistency throughout the manuscript.

*- Figure 4: It would be helpful to include n values here – for example, my understanding is that the emissions from the compression station facilities in Appalachian north came from 1 detection (n=1). I think the sample size is important to communicate here.*

Thank you for the suggestion. Figures 4, 5, S6.1 and S6.2 have all been updated to include the total number of plumes included.

-Line 255: How many measurements of pipelines were taken? Are there less large point sources because there was less observation, or are they just less prevalent in pipelines?

33/423 (from Figure 2a) of the detected plumes came from pipelines. A comparison of the detected pipeline emissions normalized by the length of pipeline observed, as done in Yu et al. 2022, is not included and we believe it more suitable for an in-depth study on exclusively pipeline emissions. From figure 4b, pipeline emissions make up 1-51% of total point source emissions from plumes ≥550 kg/hr, depending on the basin. As discussed in section 3.4, relative contribution of emissions may change with the size class of the emission rate. Nonetheless, these proportions are similar to values reported in Cusworth et al. 2022 (table 2.)

for the Permian (9-23%), Uinta (34%), and Denver-Julesburg (7-28%), despite the lower detection limit of the ANG and GAO instruments to MethaneAIR. From figure 5 (previous S6.1), we note that the total point source emissions from pipelines can vary significantly temporally, as evidenced by the lack of pipeline plumes in this size class in the Permian-Delaware from the 2021 Fall ANG and 2023 MX campaigns relative to prior campaigns in the region.

In-depth investigation on whether pipeline source are more or less prevalent in a certain emission rate size classes could be accomplished by comparing the emissions distributions curve per facility type from this study (Figure 2b) to one generated using instrument(s) with a lower detection limit (as is done in Williams et al. 2025, but this study does not delineate pipeline emissions).

*- I would include more information on the comparison of these measurements with other measurement campaigns. I think this needs to be explored more, and to comment on whether the MethaneAIR distributions of high emission point sources agree with other data sources. I also think the information provided on Page 14 could be organized into a table or figure for more effective comparison.*

Thank you for the comment. The cross-platform comparison was organized into a figure in the supplement (Fig S6.1). It has now been moved to the main body of the text as Figure 5.

Comparing the emission magnitudes in the overlapping sampled regions for plumes ≥550 kg/hr, Section 3.3, we see agreement in the emission estimates with some variation that could be due to yearly or seasonal variation and intermittency of underlying sources. From section 3.3, 2023 total point source emissions results from MethaneAIR in the Denver-Julesburg and Uinta are consistent with prior campaigns by Carbon Mapper. Results in the Appalachian-Central and Permian indicate generally decreasing emissions over time. Possible seasonal variability may mask temporal trends, as these campaigns were not all at the same time of the year.

---

## Author Response (AR2)

Dear editor,

We appreciate the opportunity to submit a revised version of our manuscript and thank the reviewer for their helpful comment.

In response to the remaining reviewer comment, we have added one paragraph and a results table to supporting information section one that details the sensitivity tests for the plume identification algorithm. We have also added language within said paragraph clarifying the criteria for valid true plumes during visual review.

Copied below is the reviewer's comment in black italics with the newly added paragraph copied as a response in blue.

We look forward to the opportunity to publish in ACP.

Sincerely,

Jack Warren, on behalf of the authors.

*I have one remaining comment/issue prior to recommending for publication. The authors now clarify that their plume detection approach hasn't been tested on single-blind controlled releases, because in those studies the source origin is known. However, they still justify the use of their hard coded values for automated plume detection as these values presumably reduce false positives. They mention that they tested over a range of values. For reproducibility and to motivate further study, can the authors include the results of this sensitivity study? Also, how can one claim they reduced false positives in absence of a truth dataset? Or, how was "truth" assessed?*

Sensitivity studies were conducted to find the combination of parameters that led to the greatest number of verifiable plumes found, with a reasonable fraction (<60%) of false positives. Plumes were considered valid if the plume was associated with identifiable infrastructure and appeared visually comparable to prior plumes detected by the platform, including those verified by controlled release testing. The size of the box used to calculate the gridded flux divergence product was tested over values ranging from 200 to 800 m. The thresholds for masking gridded DI "clumps" and XCH4 plumes were tested for values from 1 to 2.4 times the standard deviation of the scene's gridded flux or XCH4. The number of contiguous DI pixels required for a plume detection was varied from 8 - 36; the number of contiguous XCH4 pixels required for a plume detection was varied from 100 to 500. The values tested, results, and parameters chosen are shown in Table S1.1. We thoroughly tested scenes in the Permian, Appalachian, and Haynesville basins because these areas have differing surface albedos and wind speeds, and numerous visible plumes. These values will need to be reassessed for different platforms - for example MethaneSAT, which had more coarse resolution than MethaneAIR, required a separate sensitivity study and different values for these parameters.